# Developing an Abundance Index of Skipjack Tuna (*Katsuwonus pelamis*) from a Coastal Drifting Gillnet Fishery in the Southern Waters of Indonesia

**Dian Novianto [1,†], Ilham [2] , Chandara Nainggolan [2], Syarif Syamsuddin [2], Arief Efendi [2], Sugianto Halim [2], Yaser Krisnafi [2], Muhamad Handri [2], Abdul Basith [2], Yusrizal [2], Erick Nugraha [2], Suciadi Catur Nugroho [3] and Bram Setyadji [3,*,†]**

[1]   Marine Research Center, Jakarta 14430, Indonesia; dianovianto78@gmail.com
[2]   Department Fishing Technology, Faculty Fishing Technology, Fisheries University Jakarta,
      Jakarta 12520, Indonesia; ilham.jfu@kkp.go.id (I.); chandramp.nainggolan@gmail.com (C.N.);
      tigershark007@gmail.com (S.S.); arief_dkp05@yahoo.com (A.E.); Sugianto.halim@ymail.com (S.H.);
      yaser_bunda@yahoo.com (Y.K.); handrimuhammad@gmail.com (M.H.);
      abdulbasith_ppsstp63@yahoo.com (A.B.); buyung_trc@yahoo.co.id (Y.);
      nugraha_eriq1@yahoo.co.id (E.N.)
[3]   Research Institute for Tuna Fisheries, Bali 80224, Indonesia; suciadi.cn@gmail.com
*    Correspondence: bram.setyadji@gmail.com; Tel.: +62-361726201
†    These authors contributed equally to this work.

**Abstract:** Skipjack tuna is targeted by various types of fishing gear in coastal countries. Due to its resilience, it has withstood heavy fishing pressure in the past few decades. Coastal drifting gillnet fleets also mark skipjack as their main target, but it is often overlooked in terms of stock assessment. This study provides new information on an abundance index based on fishery-dependent data from 2010 to 2017. Generalized linear models (GLMs) were used to standardize the catch-per-unit-of-effort (CPUE) using year, quarter, and gross tonnage as the prediction variables. Model goodness-of-fit and model selection were based on the Akaike Information Criterion (AIC), the pseudo coefficient of determination ($R^2$), and model diagnostics with a residual analysis. The final estimation of the abundance index was calculated by least square means or marginal means. The results showed that the index was heavily influenced by the year and quarter, but it did not relate to the vessel's capacity. While the CPUE series fluctuated greatly, it showed a declining trend over the years of observation.

**Keywords:** abundance index; small scale; drifting gillnet; generalized linear model; Indian Ocean

## 1. Introduction

Skipjack tuna (*Katsuwonus pelamis*) is commercially valuable and exploited by many coastal countries along the Indian and the Pacific Oceans [1]. Since the introduction of fish aggregating devices (FADs), as well as the arrival of industrial purse seiners in the late 1980s, the catch of skipjack in the Indian Ocean has increased substantially [2,3]. The catch of skipjack tuna from Indonesian fleets accounted for 21% of the total catch in the Indian Ocean [4]. The average catch of tunas caught by Indonesian fleets in the Indian Ocean area from 2012 to 2016 was estimated to be around 150,062 tons, where the catch was dominated by skipjack tuna (51.4%), yellowfin tuna (25.9%), bigeye tuna (17.3%), and albacore (5.4%) [4].

Indonesia is currently the largest producer of skipjack tuna in the Indian Ocean [2]. However, catches of skipjack from Indonesian fleets have declined over the years. Similar trends were also reported by the French purse seine fleets [5] and Maldivian fleets [6]. Most of the skipjack tuna were

caught by purse seiners, but drifting gillnets also reported a large catch in the Indian Ocean. Drifting gillnets accounted for 20–30% of the total catch of skipjack tuna, with the main fleets operating out of Sri Lanka, the Islamic Republic of Iran, and Pakistan [2]. In Indonesia, they contributed around 17% (~13,000 ton/year) of the total national catch in the Indian Ocean region [4]. Despite their contribution, drifting gillnets are often overlooked in terms of stock assessment. Publications are scarce and limited, but include discussions about technical and economic aspects [7], catch rates (pelagic sharks) [8], and biological aspects of neritic tuna [9]. However, discussions related to the parameter population of skipjack tuna, such as catch-per-unit-of-effort (CPUE), are still limited. The access of reliable time-series catches and effort data are allegedly the main issue.

Cilacap is considered the biggest coastal gillnet fishery targeting skipjack in the southern waters of Indonesia. The fisheries were running with a consistent spatial pattern (tracking free-schooled skipjack offshore) throughout the year [9]. However, since the deployment of FAD by hand line and purse seine fleets, the fishermen have been forced to shift their operations to shore-waters. Most of the gillnet fleets are obsolete, wood-based, and they use ice mediated refrigeration. A single trip duration is usually ranged from 10 to 15 days. However, recently, some of the newly-built vessels have been equipped with in-board refrigeration systems, which can prolong the duration of fishing trips by up to 30 to 45 days/trip [10]. Recently, some of the gillnet and artisanal longline fleets were reverted to allow for hand line fishing during operation, because it is more effective in terms of productivity compared to gillnet [11–13].

CPUE is essential to stock assessment because it represents the abundance index (number or biomass) [14,15]. Ideally, it should be gathered from fishery-independent data, but since data collection is costly and often difficult, fishery-dependent data are usually used for tuna and shark CPUE assessment [16]. In the recent stock assessment on skipjack [2], the stock status was declared as not overfished and not a subject of overfishing. The stock is at the target biomass reference point, the fishing mortality is below the target, and the current catch (2016) is below its maximum sustainable yield (MSY).

The abundance indices used for the assessment were mostly from the Western Indian Ocean region, that is, European Union (EU) industrial purse seiners, Maldivian poles and lines, and Sri Lankan gillnets. However, no historical abundance indices from the Eastern Indian Ocean areas were used (e.g., from Indonesia), which would have strengthened the robustness of the previous assessments. In this study we provide new information on an abundance index based on fishery-dependent data, especially from the coastal drifting gillnet fisheries. We believe the results are valuable in terms of filling a research gap and contributing auxiliary information to assess the status of skipjack in the Indian Ocean.

## 2. Results

### 2.1. Descriptive Catch Statistics

Coastal drifting gillnet fishing targeting skipjack takes place throughout the year. August is the peak season for coastal gillnet fishing, with a highest average catch of 1908.49 ± 57.25 kg. Low season is from November to January, with average catches below 1200 kg. High standard error (SE) values were found from December to April as a result of a low sample size—less than 100 vessels/month (Figure 1). The spatial distribution of the effort was represented by $\frac{1}{4}$ degree blocks with darker and lighter colors representing, respectively, areas with more and less effort in days-at-sea. Higher efforts were concentrated in the area around 60 nm from port, while the fishing area spanned from 105–111° E. Almost all the trips were conducted inside the Indonesian Exclusive Economic Zone (EEZ) (Figure 2). Total efforts (excluding 2016) were between 2600 and 4992 days or 161 and 337 trips, with an average of 3923.57 ± 54.13 days/year. The lowest effort recorded was in 2016, and the highest was in 2012 (Table 1).

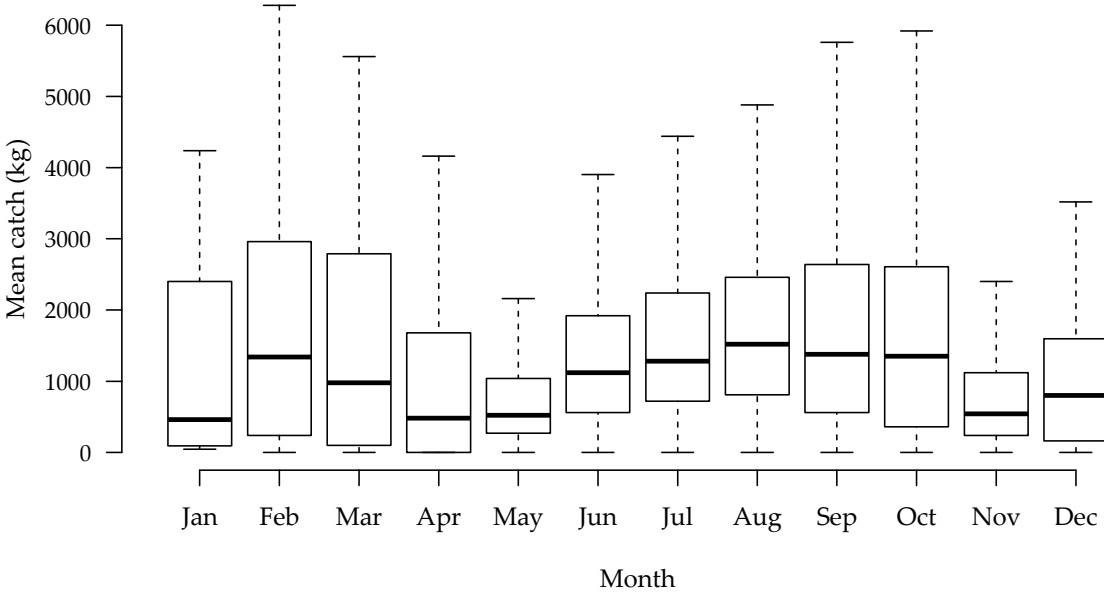

**Figure 1.** Mean monthly catch of skipjack from coastal gillnet fleet based in the Ocean Fishing Port (OFP) of Cilacap, Central Java during 2010–2017 (the bars represent standard error).

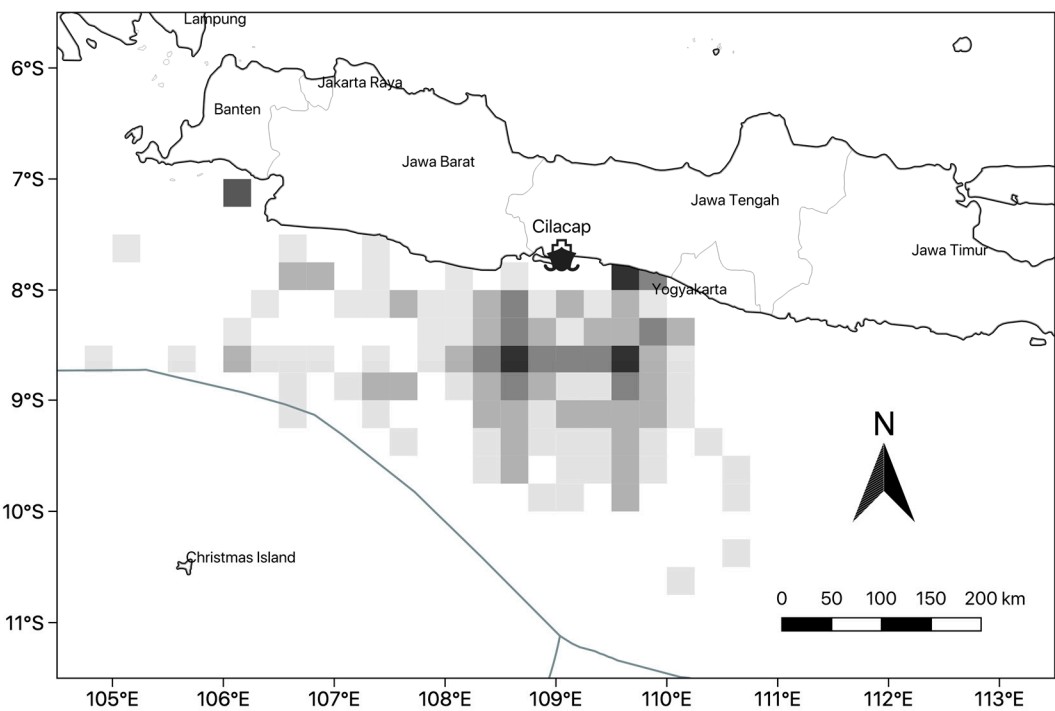

**Figure 2.** Distribution of drifting gillnet fishery efforts by $\frac{1}{4}$ degree blocks, with darker and lighter colors representing intensity (in days at sea).

**Table 1.** Trip summary of coastal gillnet fishery during 2010–2017. Results are pooled and also presented by year of observation. Operational parameters are means and standard deviations (inside parentheses). GT: gross tonnage.

| Year | Trips | Total Days-at-Sea | Mean GT | | Mean Latitude (°S) | | Mean Longitude (°E) | |
|------|-------|-------------------|---------|---|--------------------|---|---------------------|---|
| 2010 | 178 | 2680 | 24.4 | (3.4) | 8.3 | (0.3) | 109.7 | (0.0) |
| 2011 | 296 | 4437 | 24.1 | (3.6) | 8.3 | (0.4) | 109.5 | (0.7) |
| 2012 | 337 | 5008 | 24.7 | (3.6) | 7.8 | (0.1) | 109.5 | (0.7) |
| 2013 | 246 | 3866 | 24.7 | (3.4) | 7.8 | (0.1) | 109.5 | (0.5) |
| 2014 | 161 | 2600 | 24.3 | (3.7) | 7.8 | (0.1) | 109.4 | (0.8) |
| 2015 | 282 | 4405 | 24.2 | (3.6) | 8.8 | (0.4) | 108.9 | (0.7) |
| 2016 | 13 | 219 | 23.1 | (3.6) | 9.1 | (0.7) | 108.2 | (1.0) |
| 2017 | 271 | 4706 | 24.2 | (3.3) | 8.5 | (0.3) | 108.8 | (0.9) |

## 2.2. Catch-per-Unit-of-Effort Data Characteristics

The nominal CPUE was highly variable throughout the years (Figure 3). It ranged from $25.70 \pm 4.82$ kg/day to $220.28 \pm 19.04$ kg/day. The highest CPUE occurred in 2012, while the lowest was reported in 2016. There was high variation in some years, especially in 2010, 2011, and 2015.

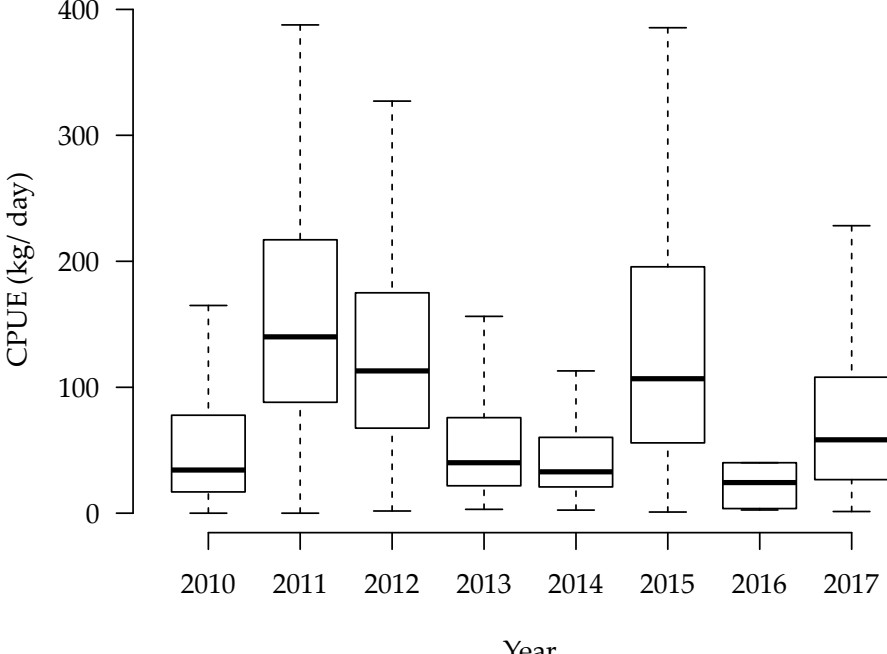

**Figure 3.** Nominal catch-per-unit-of-effort (CPUE) series (kg/day) of skipjack tuna from Indonesian drifting gillnet fishery between 2010 and 2017.

## 2.3. CPUE Standardization

Several explanatory variables tested for the skipjack CPUE standardization were significant sources of variation. Interactions were excluded to avoid overfitting. Area was dropped from the model because it provoked multicollinearity. For the final model, year had the most influential effect in the model. On the other hand, gross tonnage (GT) was not significant as shown in Table 2.

**Table 2.** Deviance table of the parameters used for the skipjack CPUE standardization using a generalized linear model (GLM) with gamma distribution. For each parameter, the degrees of freedom (Df), the deviance (Dev), the residual degrees of freedom (Resid. Df), the residual deviance (Resid. Dev), the Chi-square test statistic (Pr (>Chi)) and the significance (*p*-value) are shown.

| Parameter | Df | Deviance | Resid. Df | Resid. Dev | Pr (>Chi) | |
|---|---|---|---|---|---|---|
| (intercept only) | | | 1248 | 1261.03 | | |
| Year | 7 | 232.976 | 1241 | 1028.05 | <0.001 | *** |
| Quarter | 2 | 41.614 | 1239 | 986.44 | <0.001 | *** |
| GT | 1 | 0.200 | 1238 | 986.24 | 0.646 | |

In terms of model diagnostics, the residual analysis, including the residuals distribution along the fitted values, the QQ plots, and the residuals histograms, it was possible to detect the presence of some outliers. Residual analysis showed that the model fit the data quite well with no major outliers or trends (Figure 4).

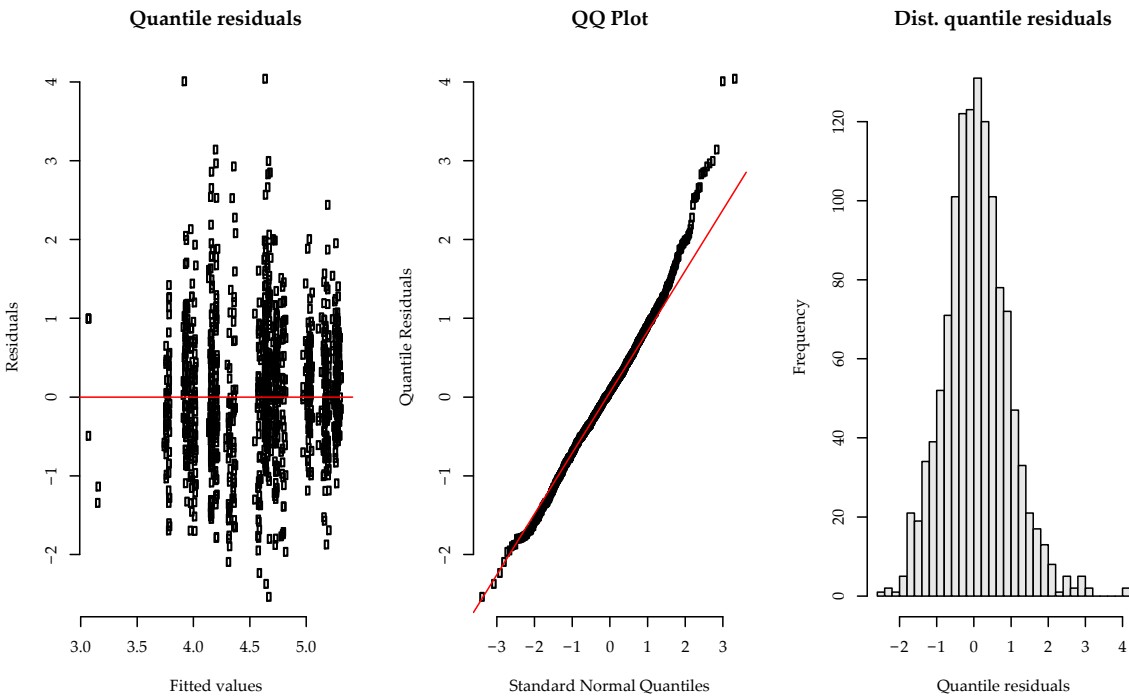

**Figure 4.** Residual analysis for the final skipjack CPUE standardization model from coastal drifting gillnet fishery between 2010 and 2017. Plots presented include the Pearson's histogram of the distribution of the residuals (**right**), the Quantile-Quantile Plot (**middle**) and the residuals along the fitted values on the log scale (**left**).

The final nominal and standardized skipjack CPUE index (kg/days-at-sea) for the coastal drifting gillnet fishery are shown in Figure 5 and Table 3. Overall, the trend was highly fluctuating and showed a declining gesture over the years of observation, especially in the last three years of the series.

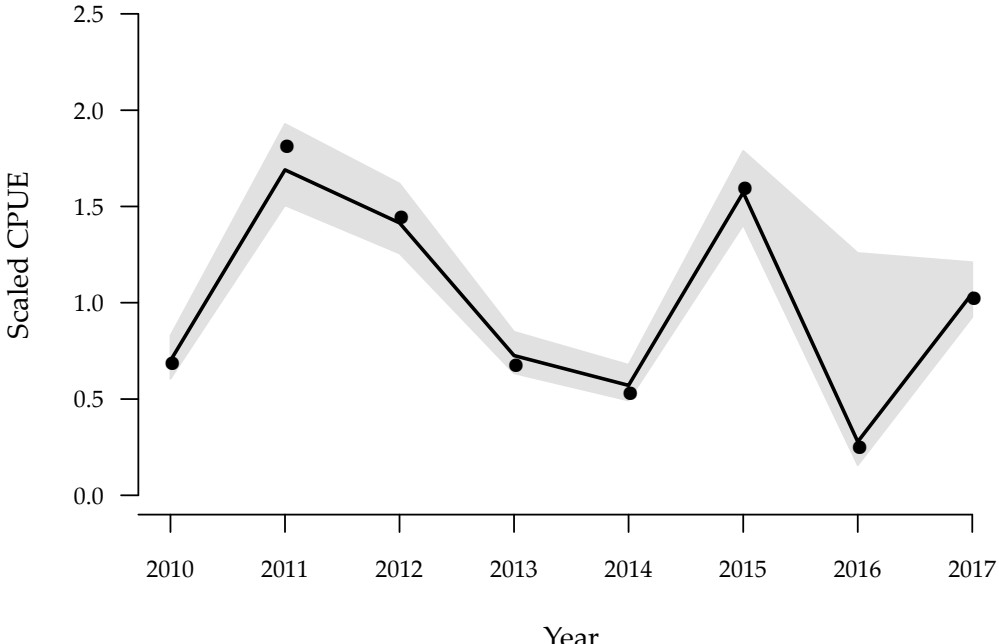

**Figure 5.** Scaled standardized CPUE series for skipjack from coastal drifting gillnet fishery using a gamma model, between 2010 and 2017. The solid lines refer to the standardized index (Std. CPUE) with the 95% confidence intervals (grey polygon), and the black dots represent the nominal CPUE series. Both series are scaled by their means.

**Table 3.** Summary of the nominal CPUE, standardized index (Std. CPUE), standard error (SE), asymptotic lower confidence interval (asymp.LCL), and asymptotic upper confidence interval (asymp.UCL) for skipjack from coastal drifting gillnet fishery.

| Year | Nominal CPUE | Std. CPUE | SE | asymp.LCL | asymp.UCL |
|------|--------------|-----------|------|-----------|-----------|
| 2010 | 59.61 | 56.64 | 4.49 | 49.01 | 67.06 |
| 2011 | 161.77 | 136.89 | 8.60 | 121.89 | 156.11 |
| 2012 | 130.30 | 114.56 | 7.38 | 101.72 | 131.11 |
| 2013 | 61.97 | 58.75 | 4.36 | 51.28 | 68.75 |
| 2014 | 48.07 | 46.19 | 3.73 | 39.87 | 54.88 |
| 2015 | 147.92 | 127.32 | 7.84 | 113.61 | 144.80 |
| 2016 | 22.90 | 22.50 | 8.94 | 12.64 | 101.89 |
| 2017 | 92.52 | 84.94 | 5.77 | 74.95 | 98.00 |

## 3. Discussion

Many fleets in coastal areas, especially in the southern parts of Java, Bali, and Nusa Tenggara are reliant on skipjack tuna as their main target species. Apart from gillnet, skipjack is also frequently caught with purse seine [17,18], hand line, and troll line [11,12], and some have also reported as a by-catch in longline fisheries [19,20]. Gillnet is a passive fishing gear, and unlike drifting longline, it does not require any bait. Technically, the skipper's knowledge of fish behavior (movement) and environmental conditions (i.e., sea current) will determine the amount of fish obtained. However, most—if not all—coastal drifting gillnet fishermen are dependent on FAD and a light attractor during their operations [7], an experience borrowed from small-scale purse seine fishery. It also made the fishing become more effective for catching skipjack, as shown by the very low proportion of zero catch per set (less than 1%). However, it is also known to have a variety of incidental by-catch species caught during operation. At least 14 species of sharks were reported during gillnet operations [8,21–24]. Hand line fishing around FAD during break (soaking time or time between gear deployment) is common practice done by gillnet crews.

Gamma distribution is suitable for modeling catch rates with very low zero catches [15] and when catch is declared in weight, instead of as individuals [16]. The final model of the skipjack CPUE index from coastal drifting gillnet fishery showed a highly fluctuated pattern, starting with a low catch rate in 2010 and then increased by threefold before gradually declining. Especially in the last three years of the series, the trend was unpredictable. However, in general, the condition of the catch rate trend is declining. This is similar to the recent condition worldwide [2]. It is also likely heavily influenced by external factors, such as government policies [25], social and religious affairs [26], and environmental conditions driven by monsoon periods [27,28].

Small-scale fisheries are often associated with low capital, low income [29–31] and are susceptible to any government policies, such as subsidized fuel. The low catch rate in 2010 was more likely affected by a 33% reduction in effort [32] because of a series of substantial increases in fuel prices since 2005 [25]. This is a natural consequence of the partial elimination of fuel subsidies by the government. It also brings a domino effect to many tuna fishing companies, which forced the reduced fishing operations in 2010 [32]. Changing fishing targets and gears is also a frequent practice in small scale fisheries. The very low catch rate of skipjack in 2016 was a result of shifting target catch to the highly valuable hairtail fish (*Trichiurus lepturus*) [33]. National fisheries statistics recorded that the production of hairtail during 2015–2016 was around 20,000 tons, double the average catch of the previous decade (2005–2016) which was around 10,000 tons (unpublished data). This phenomenon caused small-scale fleets to change their licenses and convert their gears into small longline (less than 100 m long), because in that period, fishing hairtail was economically more prominent since the fishing grounds were less than 24 nm, and produced higher catch rates (1–2 tons/trip), with an always-open market at higher prices (~$3.2/kg) than skipjack (~$1/kg). In addition, the low catch rate in 2016 was also thought to be influenced by the climate anomaly. After a strong El Niño event during 2015–2016 [34,35], during the latter half of 2016, extreme wet conditions hampered Indonesia and Australia, which is caused by the strong negative Indian Ocean Dipole (IOD) and weak La Niña [34]. This phenomenon inflicted a lower intensity of upwelling, followed by an increase in sea surface temperature (SST) and a decrease in chlorophyll-a concentration in Southern Java waters [36], which was negatively correlated with the abundance of skipjack [37,38].

Fishery-dependent data have been widely used for modeling the skipjack tuna catch rate, such as in Palabuhanratu, West Java [39] and Prigi, East Java [40]. Despite their usefulness, fishery-dependent data usually contain some observation error. It is usually related with catch abnormality in a single year (very high or very low), low effort (1–5 days) but high catch rate, and incorrect geographical information. Proper data inspection is necessary before conducting any further analysis, ignorance will lead to misjudgment or the wrong conclusion. In this study, we used the recapitulation of daily fish tickets reported to the local fisheries authority. This should be a robust data source since it was a result of the actual transactions. Unfortunately, we found out that only less than 25% was able to be analyzed. Even so, although the results presented should be considered preliminary, this study provides insights into the relationship between catch rates of skipjack and related fishing patterns. However, considering that the catch contribution from Indonesian coastal gillnet fisheries is low (around 30 tons/year), the standardized CPUE could not make a large contribution to the stock assessment of skipjack in the Indian Ocean. A joint CPUE series between fleets and countries along the Indian Ocean is encouraged to produce wider spatial coverage and increase the robustness of the assessment. Future studies involving the collection of size information and a tagging program or gene flow analysis will improve not only the abundance index itself but also our understanding of the biology and ecology of this species.

## 4. Materials and Methods

### 4.1. Gear Characteristics

Drifting gillnets gear could be scarcely found along the Southern part of Indonesia, but the largest port-based coastal gillnet fishery targeting skipjack is located in the Ocean Fishing Port (OFP) Cilacap, Central Java. It was characterized by relatively small boats with a gross tonnage around 18–30 GT. The length overall (LOA) ranged from 14 to 19 m. The length of the nets was under 2000 m (~1200 m). It consisted of 40 pieces, where every one of them was sized at 30 m (length) × 45 m (width). In terms of the nets' material, it was divided into two categories: multifilament and monofilament. However, both types were not included in the analysis, as the information was not available.

### 4.2. Fisheries Data

A total of 5288 trip data from 399 unique vessel names were collected and extracted from form of seaworthy (SL3) which is a tabulation of reported catch and effort from the fishers (fish tickets) from January 2010 to December 2017. The form was submitted to the Cilacap Ocean Fishing Port (OFP) authority right after the catch was sorted, weighted, and transported. However, as expected, because the data contained a considerable amount of errors and inconsistencies, a cleaning process was undertaken. In terms of analysis, only 30 vessels were used (7.51%). It was limited only to vessels constantly fishing throughout the years (the data from 2016 was excluded), so the dataset was reduced to only 1252 trips (23.67%). Overall, the proportion of zero catches were very low (0.23%), therefore the distribution of errors was assumed to be normal.

### 4.3. Model Selection

Catch-per-unit-of-effort was described as the total biomass of the skipjack (kg) caught by trip-basis per vessel. Since the effective fishing days were not available in the dataset, total days at sea was used instead. In this study CPUE was modeled using a generalized linear model (GLM). It could be written in matrix notation as: $g[E(Y)] = X\beta$, where $Y$ is a vector of realizations of the response variable; $E[]$ is the expectation function, $g()$ is the link function, $\beta$ is the vector of parameters, and $X$ is the design matrix of the explanatory variables. A probability distribution for $Y$ and a link function need to be selected in advance to calculate estimations of the parameters $\beta$ which represent the effects of the explanatory variables (e.g., quarter). For this approach, GLM with gamma error distribution was considered as the best approach, since the number of zero catches was not significant. The following variables from each record were considered in the model:

| | | |
|---|---|---|
| Catch | = | Total biomass of the skipjack caught per trip, stated in kilograms. This was treated as a response variable; |
| Effort | = | Time interval between departure and arrival of the vessels, ranging from 5 to 35 days. This was treated as a response variable; |
| Vessel ID | = | Categorical variable, unique identifier of each vessel; |
| Year | = | Categorical variable, range from 2010 to 2017; |
| Quarter | = | Categorical variable, represented by 1 to 4 (Quarter 1 = January–March, Quarter 2 = April–June, Quarter 3 = July–September, and Quarter 4 = October–December); |
| Capacity | = | Continuous variable, represented by gross tonnage (GT); |
| Area | = | Defined as unique value representing $\frac{1}{4}$ degree blocks. |

In order to determine which explanatory variables were to be included in the full model, the simple models were fitted with one variable at a time. The variable providing the model with the lowest residual deviance was selected first. As the second step, the model with the selected variable then received other variables, one at a time, and the model with the lowest residual deviance was again selected. This procedure continued until deviance did not decrease as new variables were added to the previous selected model.

The significance of the explanatory variables in the CPUE standardization models were assessed by likelihood ratio tests comparing each univariate model to the null model and by analyzing the deviance explained by each covariate. Goodness-of-fit and model comparison was carried out with the Akaike information criterion (AIC) [41] and the pseudo-coefficient of determination ($R^2$). Interactions were excluded to avoid overfitting, and the significant interactions were used in the analysis. Model diagnostics were carried out with a residual analysis. The final estimated index of abundance was calculated by least square means (emmeans package) [42]. For comparison purposes, these were scaled by its mean. Statistical analysis was undertaken with R Project for Statistical Computing version 3.5.0 [43], using several additional libraries, that is, nlme [44], MASS [45], and modEva [46]. The map was produced using QGIS version 2.18 [47].

## 5. Conclusions

The abundance index of skipjack tuna was heavily influenced by the year and quarter, but it did not relate with the vessel's capacity. In general, although the CPUE series was highly fluctuating, it showed a declining trend over the years of observation.

**Author Contributions:** D.N. & B.S. were the main contributors, involved in sharing ideas, designed/re-designed and performed the initial statistical models, and wrote the paper. I., C.N. also designed the experiment and analyzed the data. S.H., S.C.N., S.S., A.E., Y.K., M.H., A.B., Y., and E.N. collected and tabulated fishery data.

**Funding:** This research received no external funding.

**Acknowledgments:** We would like to thank OFP Cilacap, Central Java, Indonesia for providing fisheries data. We thank the United Nations Industrial Development Organization (UNIDO) for financial support. We would like to express appreciation to Paul McShane, Monash University, Australia for valuable advice and comments. We also gratefully acknowledge the statistical analysis advice of Rui Coelho, Portuguese Institute for the Ocean and Atmosphere.

**Conflicts of Interest:** The authors declare no conflict of interest.

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
