# Peer review of "Developing an Abundance Index of Skipjack Tuna (Katsuwonus pelamis) from a Coastal Drifting Gillnet Fishery in the Southern Waters of Indonesia"

_fishes, doi:10.3390/fishes4010010_

Round 1
Reviewer 1 Report
The second version of the paper has been improved to some extent. I think this manuscript will be acceptable after some corrections have been done. Followings are my minor two comments;
Figure 1 is a newly added but authors should carefully consider how to show mean as “bars” and standard error. Figure represented by mean as barplot and standard error as line are inappropriate. Mean should be replaced as point.
There were no information regarding fish size and migration as suggested before. If there are no information, brief future works to collect size information and to understand movement or migration to improve not only abundance indices itself but our understanding of biology and ecology of this species.
I hope these comments will be helpful.
Author Response
Please find the attached of the rebuttal letter.

Reviewer 2 Report
This revised MS has some improvement compared with the first version. However, there are still many parts need to be corrected and clarified before it can be fully accepted for publication. I suggest the authors make a major revision by taking account my comments attached to the PDF.
Some major problems are as follows:
The title needs to be changed.
The amount of catch for the drifted gill net fishery has not been mentioned in this MS. So, the reader can not understand the importance of the CPUE derived from this study. As skipjack is a highly migratory and widely distributed species in the Indian Ocean, if the local catch is only a very small portion in the Indian Ocean, the authors should make more conservative conclusion on population variation for the stock.
The Materials and Methods section needs to be moved to section 2 and more detailed definition of CPUE and model equation should be described.
The highly uncertainty of the catch and effort data may lead to a biased estimation of standardized CPUE. See the comments in Results section.
In Discussion section, how to improve the uncertainty needs to be addressed.
The conclusion needs to be more conservative.

Author Response

(The authors gave the same response as above.)

Round 2
Reviewer 1 Report
The manuscript has been revised well and much improved.
Author Response
18th January 2018
The authors would like to thank the reviewers for their specific and helpful comments.
Please find enclosed the edited manuscript in word format[file name: fishes-414269.docx]
Title: Developing an Abundance Index of Skipjack Tuna (Katsuwonus pelamis) from Coastal Drifting Gillnet Fishery in Southern Waters of Indonesia
Authors: Dian Novianto, Bram Setyadji, Ilham, Chandara Nainggolan, Suciadi Catur Nugroho, Syarif Syamsuddin, Arief Efendi, Sugianto Halim, Yaser Krisnafi, Muhamad Handri, Abdul Basith, Yusrizal, Erick Nugraha
Manuscript no: 414269
The manuscript has been improved according to the suggestions of the reviewer. The minor corrections have been fixed and some discussions have been added as per suggested by the reviewers.
We hope that our modifications render our manuscript in its current form suitable for publication in Fishes-MDPI
Yours sincerely.
On behalf of the authors.
Bram Setyadji
Reviewer 2 Report
This revised version has improved a lot in the quality. Most of my comments have been answered. However, there are still some minor corrections needed before it can be accepted for publication.
As I mentioned in the previous comments, the catch of this area is very minor (about 30 tons per year), so the standardized CPUE can not make large contribution to the stock assessment of skipjack in the Indian Ocean. The author should mention this limitation in Discussion section.
Minor comments:
Line 100, delete "have".
Line 107, "the highest".
Line 170, GT is not significant as shown in Table 2.
Line 285, "index".
Line 335, "did not".
Line 347, "the distribution of errors was assumed to be normal".
Author Response

(The authors gave the same response as above.)
